# Reduction of Amyloid Burden by Proliferated Homeostatic Microglia in *Toxoplasma gondii*-Infected Alzheimer’s Disease Model Mice

**DOI:** 10.3390/ijms22052764

**Published:** 2021-03-09

**Authors:** Ji-Hun Shin, Young Sang Hwang, Bong-Kwang Jung, Seung-Hwan Seo, Do-Won Ham, Eun-Hee Shin

**Affiliations:** 1Department of Tropical Medicine and Parasitology, Seoul National University College of Medicine, and Institute of Endemic Diseases, Seoul 03080, Korea; charisma4395@naver.com (J.-H.S.); yshwang0404@naver.com (Y.S.H.); stopsh23@naver.com (S.-H.S.); gkaehdnjs@daum.net (D.-W.H.); 2Institute of Parasitic Diseases, Korea Association of Health Promotion, Seoul 07649, Korea; mulddang@snu.ac.kr; 3Seoul National University Bundang Hospital, Seongnam 13620, Korea

**Keywords:** *Toxoplasma gondii*, chronic infection, homeostatic microglia, Alzheimer’s disease, 5XFAD mouse, disease-associated microglia, plaque-associated microglia, plaque-free microglia, lysosomal digestion

## Abstract

In this study, we confirmed that the number of resident homeostatic microglia increases during chronic *Toxoplasma gondii* infection. Given that the progression of Alzheimer’s disease (AD) worsens with the accumulation of amyloid β (Aβ) plaques, which are eliminated through microglial phagocytosis, we hypothesized that *T. gondii*-induced microglial proliferation would reduce AD progression. Therefore, we investigated the association between microglial proliferation and Aβ plaque burden using brain tissues isolated from 5XFAD AD mice (AD group) and *T. gondii*-infected AD mice (AD + Toxo group). In the AD + Toxo group, amyloid plaque burden significantly decreased compared with the AD group; conversely, homeostatic microglial proliferation, and number of plaque-associated microglia significantly increased. As most plaque-associated microglia shifted to the disease-associated microglia (DAM) phenotype in both AD and AD + Toxo groups and underwent apoptosis after the lysosomal degradation of phagocytosed Aβ plaques, this indicates that a sustained supply of homeostatic microglia is required for alleviating Aβ plaque burden. Thus, chronic *T. gondii* infection can induce microglial proliferation in the brains of mice with progressed AD; a sustained supply of homeostatic microglia is a promising prospect for AD treatment.

## 1. Introduction

Microglia are resident macrophages of the central nervous system (CNS), constitute 5–10% of total brain cells, maintain CNS homeostasis, and protect the CNS by phagocytosing pathogens [1,2,3,4]. Microglial heterogeneity, defined in relation to key regulators, markers, and drug targets, is categorized by homeostatic, pro-inflammatory, and anti-inflammatory microglial subtypes [5]. In addition, microglia play important disease-modifying roles in neurodegenerative diseases, including Alzheimer’s disease (AD), as innate immune cells of the CNS [5,6,7]. Transcriptome studies have indicated that homeostatic microglia gradually adopt a unique phagocytic disease-associated microglia (DAM) phenotype in neurodegenerative disease, chronic inflammation, and advanced aging [5]. In contrast, researchers have not yet determined the effects of microbial pathogens, which induce microglial proliferation, on AD progression using the CNS infection model; additionally, detailed studies on the effect of microbial infection in advanced AD are limited.

*Toxoplasma gondii*, a protozoan parasite that commonly infects humans and animals, chronically infects the brain, which acquires immunity against toxoplasmosis [8,9]. In the brain, *T. gondii* infection induces the expression of anti-inflammatory cytokines and negative regulators of toxoplasmic encephalitis, namely suppressor of cytokine signaling 1 (SOCS1) and Arg1, to reduce the inflammatory response [8,9]. Through a series of immunomodulatory processes, microglia increase significantly from three to nine weeks after *T. gondii* infection and slightly decrease after 9 weeks [9]. In our previous study, we demonstrated that neuroinflammation was suppressed and homeostasis restored by increasing anti-inflammatory cytokine production in the state of disease-associated inflammatory response [8,9]. However, the proliferation and role of microglia in chronic *T. gondii* infection has not been sufficiently discussed. Given that microglia continuously monitor the surrounding parenchyma to sense alterations in brain function and are involved in controlling neuronal excitability, synaptic activity, neurogenesis, and clearance of apoptotic cells in the healthy adult brain [10], it would be interesting to determine how microglia affect chronic neurodegenerative diseases, such as AD, after *T. gondii* infection.

Microglia are categorized by their phenotypes in states of health or disease [1,2,3,4,5,6,7,10]. Microglial function in the normal brain has been described as that of surveillance and limiting of the over-production of neurons, as well as excitatory synapses, through microglia-mediated phagocytosis to maintain homeostasis [2]. However, microglia are activated by acute insults and chronic disease states; reactive microglia are motile and destructive as they physically surround or target structures, such as dying cells, neurons, dendrites, blood vessels, and amyloid plaques, and therefore defend against CNS diseases [2]. In *T. gondii* infection, the microglial phenotype is significant for the disease-associated immune response [9]. *T. gondii* infection induces M1 polarization of microglia; however, at the same time, it reduces detrimental inflammatory immune responses through the induction of SOCS1, reduction of phosphorylation of signal transducer and activator of transcription 1 (pSTAT1), and the induction of anti-inflammatory cytokines [9]. In addition, the immune environment induced by *T. gondii* infection in the CNS has been determined to play a role in the inhibition of neurodegeneration and clearance of β-amyloid plaques in Tg2576 AD mice [8]. Nevertheless, little is known regarding the functional role of *T. gondi*-induced proliferation of microglia in AD pathology. In other words, the role of *T. gondi*-induced proliferation of microglia in the phagocytosis of β-amyloid plaques has not yet been reported.

The 5XFAD mouse, an experimental animal model of early-onset AD, in which β-amyloid plaques are first observed at two months of age and cognitive deficits at four to six months, is used to study the age-dependent progression of AD-like pathology [11]. *T. gondii* infection ameliorated β-amyloidosis by activating the immune response, including increased Aβ phagocytosis by Ly6C^hi^ monocytes, on day 28 after *T. gondii* infection in eight-week-old 5XFAD mice [12]. Although the microglial population increased in the brain and microglia localized to the vicinity of Aβ plaques, the role of microglia received less attention, due to their lower ex vivo phagocytic capacity, than Ly6C^hi^ monocytes [12]. However, in that study, the infection period of *T. gondii* for the ex vivo phagocytic capacity experiment was only one month, which is the early stage of infection and, thus, insufficient to evaluate the effects of infection. Given that *T. gondii* infection persists for a long time in the brain and the pathophysiology of AD is regulated over the long term, the phagocytic role of microglia in the brain has to be observed in mice older than six months with progression of AD. In that respect, we found reduced β-amyloidosis at six months after *T. gondii* infection in nine-month-old Tg2576 AD mice; however, at that time, no study on the role of microglia in ameliorating AD existed [8].

Regarding therapeutic approaches for AD, drug development is very important. Medications for AD provide temporary relief for memory loss and cognitive disorder [13] but cannot stop or reverse AD progression [13]. Therefore, the discovery of new medications and cure strategies for AD therapy is required. Activated microglia are found surrounding and phagocytosing Aβ plaques; however, microglia are also implicated in the etiology of the disease through oxidative damage and pro-inflammatory signaling [14,15]. Despite differences of opinion in defining the role of activated microglia for treating AD, targeting microglia for treating AD remains an attractive prospect [13,14,15]. The goal of a therapeutic approach targeting microglia in AD may be viewed as an attempt to return the spectrum of microglial phenotypes from the disease state (mainly synaptic pruning, reactive oxygen species generation, and inflammatory cytokine production) to the cognitively normal state (mainly, trophic support and phagocytosis) [15]. Therefore, it is important to focus on the biological role of microglial proliferation in *T. gondii*-infected AD model mice. Notably, we demonstrated that *T. gondii* infection induces M1 polarization and Th1 inflammatory responses in response to microbial infection but does not induce inflammatory responses through immune regulation during chronic infection [8,9,16]. Given that chronic *T. gondii* infection reduced amyloid burden in aged Tg2576 mice [8], the increase in microglial proliferation after *T. gondii* infection may contribute to Aβ clearance [9]. As both activation and phagocytic activity of microglia are important for disease modification in AD, this study highlights the role of microglia in reducing amyloid burden considering homeostasis and activation of microglia, close association between microglia and Aβ plaques, switch to the DAM phenotype, and phagocytic capacity and apoptosis of microglia. Simultaneously, we emphasize the importance of continuous microglial proliferation as a therapeutic target for AD.

## 2. Results

### 2.1. Microglial Proliferation and Polarization in T. gondii-Infected Mouse Brain

ME49 *T. gondii* infection induced microglial proliferation in mouse brain (Figure 1A–D). H&E-stained *T. gondii* cysts were detected during the 36-week infection period, indicating persistent infection (Figure 1A). Simultaneously, the number of Iba1-stained microglia increased from three weeks post-infection (PI) and was maintained during the 36-week infection period (Figure 1B). To investigate the proliferation of homeostatic microglia, the TMEM119 (transmembrane protein 119, a resident microglia marker) antibody was used for staining resident homeostatic microglia through the immunofluorescence assay (IFA); CD11b and Iba1 co-staining was used for activated microglia (Figure 1C,D). Mean fluorescence intensity (MFI) calculated from the IFA image stained with Iba1 using ImageJ showed an increase in the hippocampus and cortex during the 36-week infection period, with a significant increase at three and six weeks PI (*p <* 0.05, Figure 1E). By contrast, for TMEM119, MFI exhibited a significant increase at 3, 6, and 36 weeks PI (*p <* 0.05, Figure 1F) in the hippocampus and cortex during the 36-week infection period. Furthermore, the counting for Iba1 and CD11b co-stained cells showed no significant increase during the 36-week infection period (Figure 1G). Thus, most microglia proliferating after *T. gondii* infection were homeostatic microglia. Given that cell proliferation is a consequence of mitosis, the significant increase in DAPI/Iba1/Ki67-stained mitotic microglia indicates that *T. gondii* infection induces microglial proliferation (yellow arrow head and Iba1/Ki67-co-stained cells/mm^2^ brain tissue; *p <* 0.05, Figure 1H).

To investigate genetic changes that induce microglia proliferation, microarray analysis was performed for genes encoding microglial trophic factors (*Il1β*, *Tnfα*, *Mcsf*, and *NFKB1*), microglia markers (*Iba1*), resident homeostatic microglial markers (*P2ry13*, *Cx3cr1*, and *Tmem119*), and polarization markers (*Ifnγ*, *Il4*, *Cd86*, and *Cd206*) at 3, 6, 12, and 36 weeks PI (Figure 1I). Genes encoding microglial trophic factors and resident homeostatic markers continuously increased, except for the decrease in *Mcsf* at 36 weeks, during chronic *T. gondii* infection (Figure 1I). At this time, the direction of microglial polarization was not M2 (*Il4* and *Cd206*) but M1 (*Ifnγ* and *Cd86*). Our findings revealed that chronic *T. gondii* infection in the brain induced M1-type activation of microglia through the continuous proliferation of homeostatic microglia. Importantly, homeostatic microglial proliferation was maintained during chronic *T. gondii* infection.

### 2.2. Amyloid-β Plaque Reduction and Microglial Proliferation in T. gondii-Infected 5XFAD Mouse Brain

5XFAD mice were used to investigate the proliferation and activation of microglia after *T. gondii* infection. *T. gondii* cysts (black arrows in H&E and Congo red staining) were found around Aβ plaques (blue arrow head in Congo red staining) in the brain tissue at 40 weeks PI (Figure 2A). *T. gondii*-infected (AD + Toxo group) 5XFAD mice exhibited a reduction in Aβ plaque burden in the hippocampus and cortex compared with non-infected mice (AD group). Moreover, the number of acidophilic neurons, which indicate a neuropathological state in layer V of the cortex, decreased in the AD + Toxo group compared with the AD group. (Figure 2B). The number of dense core plaques stained with Congo red dye significantly decreased in the hippocampus and cortex of the AD + Toxo group compared with the AD group (*p* < 0.05, Figure 2C). Evaluating the amount of Aβ1–42 in brain tissue lysate using the amyloid β1–42 assay kit (ELISA) revealed that the amount of Aβ1–42 was significantly reduced in the AD + Toxo group (67.27 ± 18.73 pg/mL) compared with the AD group (127.5 ± 6.651 pg/mL) (*p* < 0.05, Figure 2D). Given the importance of microglia in clearing Aβ plaques, we analyzed microglial accumulation around Aβ plaques (Figure 2E). Microglial cells (DAB-stained, dark brown) accumulated around amyloid plaques in both AD and AD + Toxo groups; however, the number of microglia around the Aβ plaque was higher in the AD + Toxo group than in the AD group (Figure 2E). To compare the degree of microglial proliferation in AD and AD + Toxo groups, the hippocampus and cortex were stained using the Iba-1 Ab (red). The degree of microglial proliferation was greater in the AD + Toxo group than in the AD group (Figure 2F). To evaluate this difference numerically, when MFI of Iba-1 staining in the AD group was defined as numerically one-fold, the fold change of MFI in the AD + Toxo group was significantly increased in both the hippocampus and cortex (*p* < 0.05, Figure 2F). Similarly, resident homeostatic microglial cells stained with the TMEM119 Ab were significantly increased in the AD + Toxo group in both the hippocampus and cortex compared with the AD group (*p* < 0.05, Figure 2G). To investigate molecular signals that induce the proliferation of homeostatic microglia in the AD + Toxo group, microglial trophic factors (IL-1β and TNF-α) and inducers (IFN-γ and IL-4) for polarization of M1 and M2 microglia were examined using ELISA and microarray analysis (Figure 2H,I). Our data revealed that the proliferation of homeostatic microglia in the AD + Toxo group correlated with a significant increase in IL-1β and TNF-α levels and in the expression of trophic factors (Il1β, *Tnfα*, *Mcsf*, and *NFKB1*) (*p* < 0.05, Figure 2H,I). Thus, polarization of proliferated microglia can be inferred from the expression of M1- and M2-inducing cytokines (IFN-γ and IL-4). Moreover, the expression of M1-inducing factors (IFN-γ at the protein level and *Ifnγ* and *Cd86* at the gene level) was stronger than that for M2-inducing factors (IL-4 at the protein level and *Il4* and *Cd206* at the gene level) when comparing the AD + Toxo group to the AD group (*p* < 0.05, Figure 2H,I). Homeostatic microglia markers, namely P2ry13, Cx3cr1, and Tmem119, were found to increase in the AD + Toxo group compared with the AD group (Figure 2I). Thus, the *T. gondii*-mediated increase in microglial proliferation and polarization may be important for reducing Aβ plaque burden.

### 2.3. Plaque-Associated Patterns of Microglia and Ly6C^+^ Monocytes

To investigate the plaque-associated pattern of microglia responsible for plaque clearance, the distribution of plaque-associated microglia and plaque-free microglia was evaluated by co-staining with methoxy-XO4 (a fluorescent probe for detecting plaques) and Iba1 Ab (Figure 3A). Microglia in close contact of an amyloid plaque, such as within a 30-μm radius, were termed plaque-associated microglia and those at more than a 30-μm radius were termed plaque-free microglia (Figure 3A). Given that the morphology of the dense core amyloid plaque stained with methoxy-XO4 is large and unspecified (displayed as a white mass), the number of microglia (red fluorescence) in close contact with the amyloid plaque border was higher in the AD + Toxo group than the AD group (Figure 3A). To compare the number of plaque-associated or plaque-free microglia, we collected 60 amyloid plaques from the brain ((10 randomly selected plaques per mouse) × 6 mice per group) and counted microglia around plaques (Figure 3A). The numbers of both plaque-associated and plaque-free microglia were significantly increased in the AD + Toxo group compared with the AD group, indicating migration of newly proliferated homeostatic microglia from the far side (plaque-free pattern) to the near side of the plaque (plaque-associated pattern) (*p* < 0.05, Figure 3A). To confirm this, resident homeostatic microglia surrounding amyloid plaques were analyzed by co-staining with Iba1 (red), TMEM119 (green), and methoxy-XO4 (blue). Few TMEM119-stained microglia surrounded amyloid plaques in the AD group, whereas several TMEM119-expressing microglia (yellow) were seen in the AD + Toxo group (Figure 3B). When counting the number of TMEM119-expressing microglia around amyloid plaques, the number of plaque-associated homeostatic microglia was significantly increased in the AD + Toxo group compared with the AD group (*p* < 0.05, Figure 3B). Moreover, because *T. gondii* infection can recruit Ly6C^+^ monocytes, which inhibit amyloidosis, we evaluated the accumulation of Ly6C^+^ monocytes around amyloid plaques (Figure 3C). Few Ly6C^+^ cells (green) accumulated around amyloid plaques in both AD- and AD + Toxo groups; accordingly, the number of plaque-associated Ly6C^+^ cells exhibited no difference between AD- and AD + Toxo groups (Figure 3C). Thus, the ablation of amyloid plaques in the brain with AD progression may proceed through microglia but not Ly6C^+^ monocytes.

### 2.4. Switching of Homeostatic Microglia to the DAM Phenotype Followed by Phagocytosis of Microglia for Plaque Clearance

In AD, homeostatic microglia are activated to clear plaques. As the presence of polarized DAM microglia is important for clearing amyloid plaques, the expression of DAM-related genes was investigated in AD and AD + Toxo groups and compared with that in uninfected wild-type mice (Figure 4A). In comparison to the AD group, in the AD + Toxo group gene expression for the overall DAM phenotype, including *Apoe*, *Ax1*, *Clec7a*, *Cst7*, and *Trem2*, was reduced. By contrast, the expression of the homeostatic microglia gene *Tmem119* increased (Figure 4A). Microarray gene expression and quantitative gene expression analysis of DAM (*Cst7*) and homeostatic (*Tmem119*) markers revealed that *Cst7* was predominantly expressed in the AD group and *Tmem119* in the AD + Toxo group (*p* < 0.05, Figure 4A,B). This result suggested that most microglia in the AD group were converted to the DAM phenotype, whereas in the AD + Toxo group, many homeostatic microglia were newly proliferated although they were converting to the DAM phenotype. To visually check switching to DAM phenotype, microglia were co-stained with Alexa 568-conjugated Iba1 (red) and Alexa 488-conjugated TREM2 around the methoxy-XO4-labeled Aβ plaque (Figure 4C). Colabeling of Iba1 and TREM2 (shown in yellow) can be seen around the methoxy-XO4-labeled Aβ plaque; the number of colabeled microglia surrounding this Aβ plaque was higher in the AD + Toxo group compared with the AD group (Figure 4C). Thus, newly proliferated homeostatic microglia in the AD + Toxo group were converted to the DAM phenotype around the Aβ plaque. Moreover, another DAM-specific microglia marker, namely lipoprotein lipase (LPL), was expressed in microglial cells around the Aβ plaque in both AD and AD + Toxo groups (green; Figure 4D). Microglia co-stained with Alexa 488-conjugated LPL (green) and Alexa 568-conjugated Iba-1 (red) are shown in yellow; most plaque-associated microglia switched to the DAM phenotype in both AD and AD + Toxo groups (Figure 4D). However, given that the sustained proliferation of homeostatic microglia appeared in the AD + Toxo group but not the AD group, the pool of microglia capable of removing plaque was sufficient in the AD + Toxo group but not the AD group. To define the role of microglia recruited around the Aβ plaque, we investigated lysosomal degradation of the Aβ plaque by evaluating the colocalization of Iba1 (red) and lysosomal-associated membrane glycoprotein 1 (LAMP1; green) (Figure 4E). Iba1 and LAMP1 colocalization is shown in yellow; lysosomal degradation and clearance of Aβ plaque was indicated by the closed association between LAMP1 and the Aβ plaque. Iba1-positive microglia associated with the Aβ plaque (white) expressing LAMP1 indicated lysosomal degradation, which can be seen as internalized puncta. In this figure, two amyloid β plaques, which were fragmented through lysosomal degradation within the microglia, are represented as “a” and “b” (Figure 4E). Enlarged images (“a” and “b”) show microglia with fragmented amyloid β plaques. The degree of lysosomal degradation of the Aβ plaque, which is indicated by internalized puncta (yellow arrow) per microglial cell, was seen in plaque-associated microglia of both AD + Toxo and AD groups (Figure 4F).

### 2.5. Plaque-Associated Microglia Apoptosis after Phagocytosing the Aβ Plaque

The phenomena that microglia shifted to the DAM phenotype and did phagocytosing Aβ plaques may induce fatal cell changes, such as cell death, in microglia. We investigated whether microglia undergo TUNEL^+^-apoptotic cell death after phagocytosing Aβ plaques (Figure 5). TUNEL^+^-cells are shown in green and microglial nuclei in blue (DAPI). Co-staining with TUNEL and DAPI is shown in light blue/white (arrow) and was detected through the fragmented DNA morphology in the nucleus (Figure 5). The ratio of apoptotic microglia among plaque-associated microglia was almost 80% in both AD and AD + Toxo groups (Figure 5). Thus, plaque-associated microglia in both AD and AD + Toxo groups are apoptosed after phagocytosing Aβ plaques.

## 3. Discussion

Therapies for AD include targeting Aβ plaque formation, neurofibrillary tangle formation, and neuroinflammation [17]. However, based on results from investigating mechanism-based treatments for AD, the questions of which of the targeted processes are critical for disease progression and how best to inhibit AD remain controversial [17]. Rational translational research reveals that molecular targeting for treating AD includes the use of inhibitors against amyloid aggregation, β-secretase, and γ-secretase [13,17]. In addition, antibodies against amyloid (passive immunization); inhibitors of glycogen synthase kinase (GSK)3β, which phosphorylates tau; and nonsteroidal anti-inflammatory drugs (NSAIDs) targeting neuroinflammation can be used as AD treatment [13,17]. However, treatment results have been unsatisfactory [17]. In recent years, embryonic stem cells, brain-derived neural stem cells, and induced pluripotent stem cells have been used as cell therapy for AD treatment; however, problems persist, including the requirement for a neurosurgical procedure, immunosuppression, and tumor formation [18]. As a novel therapeutic strategy for AD, a nano-based drug delivery system was suggested [19]. Despite extensive efforts, treating AD clinically has remained a challenge. The important thing is to consider the multifactorial character of AD, and development of a single drug is unlikely to lead to universal AD therapy [19]. In sum, the development of alternative therapies is needed in order to suppress all the complications of AD by anticipating the intrinsic role of cells that can function for a long time in the brain [19,20,21].

Glial targets for effective therapies of AD were introduced as an implicative strategy for AD therapy [20]. However, glial-mediated inflammation is a “double-edged sword”, performing both detrimental and beneficial functions in AD [20,21]. In other words, although it is still debatable whether the glial-mediated inflammatory response in AD is a consequence or cause of neurodegeneration, regarding immune surveillance, microglia, as resident macrophages, can preferentially provide the beneficial functions of an immunological first defense against invading pathogens and other types of brain injury [20,21]. Studies have demonstrated the presence of activated microglia at sites of Aβ deposition, suggesting that glial cells interact with Aβ plaques and regulate plaque levels in the brain [20,22,23]. A recent study suggested that most plaque-associated myeloid cells responsible for Aβ plaque clearance are resident microglia [24].

When AD mice were induced to overexpress IL-1β in the hippocampus through adenoviral transduction, microglia number increased and microglial capacity to facilitate Aβ plaque clearance also increased [23]. Similarly, our findings revealed that *T. gondii* infection induced continuous proliferation of microglia in AD mice, resulting in Aβ plaque clearance. Despite large-scale microglial proliferation, we found no evidence that microglia proliferation increased the number of pyramidal neurons associated with neurodegeneration in the brain tissue of the AD + Toxo group. Inflammation may be the key neuropathological event leading to neurodegeneration in AD [20]. Therefore, the activation of glia, microglia, and astrocytes plays an important role in inducing the inflammatory signaling pathway during neurodegeneration [20]. By contrast, although *T. gondii* infection induced M1 polarization and microglial proliferation during 12 weeks of chronic brain infection, significant induction of inflammatory responses was not observed [9]. In this study, microglial proliferation was visualized histologically during the 36-week chronic *T. gondii* infection period, which was molecularly supported by Ki67^+^-mitosis and trophic factor gene expression, indicating the presence of newly dividing homeostatic microglia. Regarding the inflammatory response, although we have not analyzed inflammatory cytokines and inflammatory immune responses, results from our previous study revealed that the increase of SOCS1 and Arginase1 and the decrease of pSTAT1 and nitric oxide were negative regulators of toxoplasmic encephalitis [9]. Therefore, *T. gondii* infection is a new model that proves the beneficial effect of microglia on AD progression.

TREM2 expression in plaque-associated microglia is required for microglia-mediated Aβ phagocytosis [21]. In this study, the appearance of TREM2-positive microglia around Aβ plaques and the increase in lysosomal digestion of Aβ plaques were related to Aβ clearance in the AD + Toxo group. However, because the fate of DAM-switched microglia interacting with Aβ was cell death, the presence of stimuli, such as *T. gondii*, which continuously influence the proliferation of homeostatic microglia, is an important factor for Aβ clearance, as seen for the AD + Toxo group. In other words, it seems obvious that an increased number of homeostatic microglia underwent cell death after migrating from the periphery to plaque border. Our results are supported by the microglia turnover rate (newly appearing cells after disappearance of microglial cells), which increased in AD mice in the presence of amyloid lesions compared with wild-type mice [25]. A highlight of our study was that microglial proliferation was followed up for 36 weeks after *T. gondii* infection. Considering that chronic *T. gondii* infection is defined from approximately eight to nine weeks after infection and maintains the infection state over a lifelong period, our study is significant in that proliferation kinetics of microglia over a long period after infection was evaluated [9,12,26]. Most proliferated microglia were TMEM119^+^/Iba-1^+^-stained homeostatic microglia, and both plaque-associated and plaque-free microglia increased in the AD + Toxo group compared with the AD group. In AD mice, microglia migrate to amyloid plaques, phagocytose the amyloid plaques, and undergo apoptosis after lysosomal degradation of Aβ [20,27,28,29]. This process of microglia participating in Aβ degradation is also explained by microglial activation, which is triggered through highly expressed pattern recognition receptors, including CD14, CD36, and toll-like receptors on microglia surface [20,28]. In this study, microglia functioned as neuroprotective cells, clearing the pathological Aβ aggregates through lysosomal degradation. This suggestion was also supported by previous our data obtained from behavioral studies using Morris’ water maze, where AD mice infected with *T. gondii* (AD + Toxo group) exhibited amelioration of symptoms of loss of cognitive function. By contrast, Green and colleagues reported that microglial depletion impairs parenchymal plaque development []. When microglia, which depend on colony-stimulating factor 1 receptor (CSF1R) signaling for survival, were treated with a CSF1R inhibitor, their population was depleted and plaques failed to form in the parenchymal space [30]. The authors asserted that microglia play a role in the onset and development of AD pathology [30]. However, considering that microglia maintain homeostasis in the brain and play various roles, such as adult neurogenesis and neuronal circuit formation [2,3,10,31], it is reasonable to think that microglial depletion plays a role in eliminating the non-resolving inflammatory response during AD pathogenesis. However, whether this depletion is sufficient to prevent sustainably AD progression is debatable. In this context, our study highlights the role of microglia in inhibiting AD progression. In other words, our findings suggest that the characteristics of immunity induced by *T. gondii* infection, namely simultaneous increase in homeostatic microglia and suppression of inflammatory immunity, are important. In the brain, this immune environment induces an increase in Aβ phagocytosis of microglia while suppressing the non-resolving inflammatory response.

Regarding clearance of Aβ in *T. gondii*-infected 5XFAD mice, Ly6C^hi^ monocytes were shown to be associated with increased phagocytosis and degradation of soluble Aβ [12]. However, in that study, Iba1-labeled microglia but not Ly6C^hi^ monocytes interacted more closely with plaques; histological analysis of *T. gondii*-infected brains also exhibited proliferation and activation of resident microglia [12]. Nevertheless, that study concluded that Ly6C^hi^ monocytes had more phagocytic capacity than microglia when evaluated for their ability to specifically phagocytose Aβ_42_ in an ex vivo phagocytosis assay and insisted that chronic *T. gondii* infection enhances β-amyloid phagocytosis and clearance by recruited monocytes [12]. Comparing that study with our study, the biggest difference is the infection period of *T. gondii*. In the above study, 5XFAD mice were sacrificed at eight weeks after infecting eight-week-old mice with *T. gondii*, while in this study, mice were sacrificed at 40 weeks after infecting eight-week-old mice with *T. gondii*. With the passage of *T. gondii* infection time, the early inflammatory immune response is gradually converted to an anti-inflammatory immune response, and the acute *T. gondii* infection becomes a chronic infection [8,9]. The early inflammatory response of *T. gondii* in the brain results in the infiltration of Ly6C^+^ monocytes in the brain; however, chronic *T. gondii* infection exhibited an increase in microglial population [12,16,32,33]. Therefore, in this study, the interaction between microglia and Aβ and the reduction of amyloid burden in chronic *T. gondii* infection suggests that microglia are effectively inhibitors of Aβ deposition.

Taken together, *T. gondii* infection induced the proliferation of homeostatic microglia during chronic infection, and plaque-associated microglia in *T. gondii*-infected AD mice were significantly increased compared with those in uninfected AD mice. Considering that plaque-associated microglia undergo apoptosis after phagocytosing Aβ plaques, the continuous increase of microglia during *T. gondii* infection may be an important factor in reducing Aβ deposition. Notably, results from our previous study revealed that neuroinflammation and neurodegeneration were not induced during chronic *T. gondii* infection even after microglial proliferation. Therefore, the reduction in neuroinflammation should be considered when considering using microglia as a target for AD treatment. In this study, we demonstrated the potential of *T. gondii* in maintaining long-term microglial proliferation in the brain. Further studies should determine what factors can proliferate microglia in *T. gondii* itself.

## 4. Materials and Methods

### 4.1. T. gondii Infection and Experimental Design

For infecting 5XFAD mice, *T. gondii* ME49 cysts were obtained from brain tissues of C57BL/6 mice (Orient Bio Animal Center, Seongnam, South Korea) infected with 10 cysts. Seven-week-old C57BL/6 mice were infected with ten cysts for the histopathological examination and microarray analysis of the brain at 0, 3, 6, 12, and 36 weeks after infection (*n* = 2–4 per group). Eight-week-old 5XFAD mice were orally infected with ten cysts and euthanized using CO_2_ asphyxiation for histopathological examination of the brain at ten months after infection (*n* = 6).

### 4.2. 5XFAD Transgenic Mice

5XFAD mice (The Jackson Laboratory, Bar Harbor, ME, USA) were kindly provided by Dr. Inhee Mook-Jung at Seoul National University and bred by mating male mice with wild-type females (C57BL6/SJL). Genotyping for confirming the presence of human *APP* and *PSEN1* transgenes was performed through PCR analysis of tail genomic DNA. PCR products (350 bp for *APP* and 608 bp for *PSEN1*) were analyzed using 1% agarose gel electrophoresis and detected through ethidium bromide staining. Primer sequences are listed in the Appendix A.

### 4.3. Ethics Statement

All animal experiments were approved by the Institutional Animal Care and Use Committee at Seoul National University (Permit Number: SNU-110315-5). Mice were maintained in an animal facility according to the standards of the Animal Protection Act and the Laboratory Animal Act in Korea. Mice experiments were performed according to global standards, such as those established by the Association for Assessment and Accreditation of Laboratory Animal Care International. All efforts were made to minimize animal suffering (Approved Number: SNUIBC-R110302-1-1).

### 4.4. Hematoxylin and Eosin Staining and Congo Red Staining of Brain Tissue

Brain tissues were fixed in 10% formalin and embedded in paraffin after dehydration through an ethanol gradient. Brain tissues, which were coronally sectioned at 10-µm thick, were stained with Harris’ hematoxylin and eosin (H&E) to detect *T. gondii* tissue cysts and neuronal degeneration, indicated by acidophilic neurons in layer V of the cerebral cortex (*n* = 6 per group). Observations were performed using light microscopy (CKX 41, Olympus, Tokyo, Japan) and a color digital camera (DP72, Olympus). For Congo red staining, sectioned brain tissues were incubated in 0.4% aqueous Congo red solution (Sigma-Aldrich, St Louis, MO) for 10 min at room temperature (RT), counterstained in hematoxylin for 10 min, and dipped in acid alcohol (1% HCl in EtOH) for differentiation. Finally, after the sections were dehydrated and mounted, dense core plaques in the cortex and hippocampus were counted using a color digital camera attached to a light microscope and evaluated using ImageJ (Version 1.45, National Institute of Health, Bethesda, MD, USA).

### 4.5. Immunohistochemistry

Immunohistochemistry (IHC) was performed after Congo red staining. Briefly, tissue sections were deparaffinized in xylene, rehydrated in a graded series of ethanol, and rinsed with distilled water. After antigen retrieval, endogenous peroxidase activity was blocked using H_2_O_2_ in blocking buffer (1% fetal bovine serum in PBS) for 30 min. Then the slides were incubated with rabbit anti-mouse Iba-1 (Wako, VA, USA) as the primary antibody (Ab) and UltraMap anti-Rb horseradish peroxidase (HRP) (Ventana Medical Systems, Tucson, AZ, USA) as the secondary Ab. The ChromoMap DAB detection kit (Ventana Medical System) was used for detecting the DAB signal and hematoxylin was used for counterstaining. Slides were observed using a light microscope equipped with a color digital camera.

### 4.6. Immunofluorescence Staining and Quantitative Analysis

Brain sections with antigen retrieval were permeabilized with 0.5% Triton X-100 in phosphate-buffered saline (PBS) for 10 min at RT and blocked in 2% bovine serum albumin (BSA)/PBS or 10% donkey serum for double-staining. Antibodies used for immunofluorescence (IF) staining are listed in the Appendix A. Samples were stained with the corresponding secondary Ab and nuclei were stained with 4′,6-diamino-2-phenylindole, dihydrochloride (DAPI; Sigma-Aldrich, St. Louis, MO, USA). For detecting amyloid plaque, methoxy-X04 (Abcam, Pittsburgh, PA, USA) staining was performed before DAPI counterstaining. Tissue sections were stained with 100-μM methoxy-X04 in 40% ethanol (adjusted to pH 10 with 0.1 N NaOH) for 10 min and then incubated with 0.2% NaOH in 80% ethanol for 2 min. Immunostained slides were observed using fluorescence microscopy (Leica DMI6000B, Wetzlar, Germany). For quantifying area covered by microglia in the cortex and hippocampus, slides stained with Iba1 and TMEM119 Abs were used for calculating mean fluorescence intensity (MFI) using ImageJ in four identical regions (non-overlapped images at 20× magnification) captured by fluorescence microscopy (*n* = 5–6 per group). For counting immunolabeled cells, the number of cells per defined tissue area was counted at 40× and 20× magnifications. Plaque-associated cells (microglia and Ly6C^+^ monocytes) included cells at distances <30 µm from the center of the amyloid plaque. Plaque-free cells included cells at distances >30 µm from the center of the amyloid plaque. For counting the number of cells around plaques, microglia were selected based on DAPI-labeled nuclei. The result was presented as the total number of immunolabeled cells around plaques.

### 4.7. ELISA for Amyloid β (1–42) Quantification

The Amyloid β (1–42) Assay Kit (#27711, IBL, Tokyo, Japan) was used to measure amounts of Aβ42 (*n* = 7 per group). The test sample included 100 µL brain lysate, and all procedures for the assay were performed according to the manufacturer’s guidance. Reagents were prepared at RT approximately 30 min before use.

### 4.8. Multiplex Cytokine Immunoassay

Expression levels of IL-1β, TNF-α, IFN-γ, and IL-4 in 5XFAD mouse brains were examined using multiplex immunoassay (Bio-Plex mouse cytokine assay kit, Bio-Rad Laboratories, Hercules, CA, USA) (*n* = 5–6 per group). Brain tissues were lysed using the MicroRotofor™ Cell Lysis Kit (Bio-Rad Laboratories), and proteins in the homogenate were quantified using the bicinchoninic acid (BCA) assay kit (Pierce Biotechnology, Inc., Rockford, IL, USA). The assay was performed according to the manufacturer’s instructions and results were analyzed using the Bio-Plex Manager Software and Bio-Plex Data Pro™ software.

### 4.9. qRT-PCR

Total RNA from brain tissue was isolated using the HiGene Total RNA Prep Kit (BIOFACT, Daejeon, Korea) according to the manufacturer’s protocol and reverse-transcribed using the RT-PCR premix kit (Elpis Biotech Inc., Daejeon, Korea). Quantitative real-time PCR was performed using CFX96 (Bio-Rad) and SYBR green (Enzynomics™, Daejeon, Korea) (*n* = 4–5 per group). Primer sequences used for real-time PCR are presented in the Appendix A.

### 4.10. Microarray

Total RNA from brain tissue was extracted and pooled for microarray analysis (*n* = 3), which was performed using the Illumina MouseRef-8 v2 Expression BeadChip array (Illumina, Inc., San Diego, CA, USA) by Macrogen Inc. (Seoul, Korea). Export processing and analysis of arrayed data were performed using Illumina GenomeStudio v2011.1 (Gene Expression Module v1.9.0), and data were analyzed with the R v. 2.15.1 statistical software. Hierarchical cluster analysis was performed using Permute Matrix. Heat maps were created using the Excel Spreadsheet Software (Microsoft Corporation, Redmond, WA, USA) with conditional formatting. Gene expression rates in AD and AD + Toxo groups were compared with that in the wild-type (WT) group. Positive correlations are depicted in yellow (increased expression) and negative (decreased expression) in blue. The color scale of the heat map represents the relative minimum (−3) and maximum (+3) values of each gene.

### 4.11. TUNEL Staining for Plaque-Associated Apoptotic Microglia

TUNEL staining was performed on paraffin-embedded sections using the Cell Meter™ Fixed Cell and Tissue TUNEL Apoptosis Assay Kit (AAT Bio., Sunnyvale, CA, USA) according to the manufacturer’s protocol. Briefly, sections were deparaffinized, rehydrated, and immersed in 4% paraformaldehyde for 20 min at RT. Then, sections were incubated in 20 μg/mL proteinase K for 10 min and fixed with 4% paraformaldehyde for 20 min. Subsequently, sections were incubated with 25 μL TUNEL reaction mixture for 60 min and immunostained with Iba1 Ab (Abcam) as the primary Ab and anti-goat Alexa Fluor 647 (Invitrogen, Carlsbad, CA, USA) as the secondary Ab. Finally, DAPI-stained sections were mounted. TUNEL positive (apoptotic) cells were observed using fluorescence microscopy at 40× magnification (*n* = 6 per group).

### 4.12. Statistics

All statistical analyses were performed using the GraphPad Prism 5 software (GraphPad, La Jolla, CA, USA). Data are presented as the mean ± standard error of the mean. For MFI fold change data in Figure 1, one-way analysis of variance (ANOVA) followed by Dunnett’s multiple comparisons test was used for statistical evaluation. To compare experimental groups, namely WT, AD, and AD + Toxo, one-way ANOVA followed by Tukey’s multiple-comparison test was performed. Significant differences for two groups were assessed using Student’s *t*-test with Welch’s correction. An asterisk (*) indicates a significant difference compared with the control (*p* < 0.05) and a sharp (#) indicates a significant difference among experimental groups (*p* < 0.05).

## Figures and Tables

**Figure 1 ijms-22-02764-f001:**
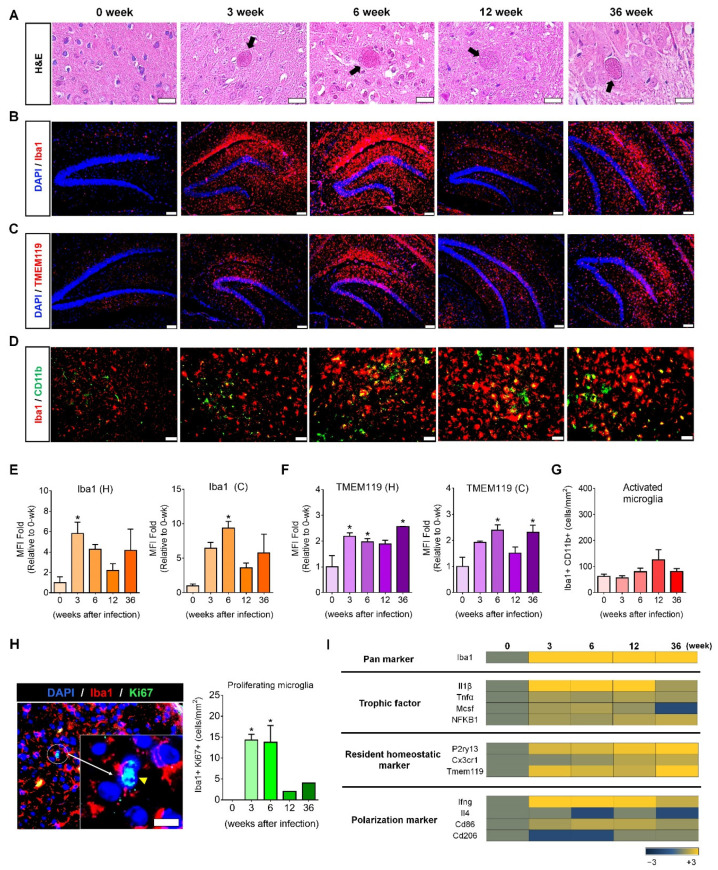
Proliferation and activation of resident homeostatic microglia over 36 weeks after *Toxoplasma gondii* infection. (**A**) *T. gondii* infection was confirmed by the presence of cysts in brain tissue (H&E staining; scale bar, 20 µm). (**B**) Microglia in the hippocampal formation were stained with Iba1 (red; scale bar, 100 µm). (**C**) Microglia in the hippocampal formation were stained with TMEM119 (red; scale bar, 100 µm). (**D**) Activated microglia were co-stained with CD11b and Iba1 (green and red, respectively; scale bar, 50 µm). (**E**,**F**) Mean fluorescence intensity (MFI) was calculated from fluorescence-stained images (**B**,**C**) using ImageJ. The fold changes of MFI at 3, 6, 12, and 36 weeks PI were compared with those of the control (0 weeks). H, hippocampus; C, cortex. (**G**) The number of activated microglia (Iba1^+^/CD11b^+^) was designated by cell number per mm^2^ of the brain tissue. (**H**) Ki67-stained proliferating microglia. The yellow arrow head shows microglial cells in the mitotic phase (sky blue) co-stained with DAPI (blue) and Ki67 (green), and the number of proliferating microglia (Iba1^+^/Ki67^+^) was designated by cell number/mm^2^ brain tissue. Scale bar; 20 µm. Quantification of MFI intensity and co-stained cell counting; two brain sections per mouse. (**I**) Microarray analysis of genes encoding trophic factors, homeostatic markers, and M1 and M2 markers in *T. gondii*-infected brain. Data are represented as the mean ± SEM. * Statistical significance compared with the control (* *p* < 0.05).

**Figure 2 ijms-22-02764-f002:**
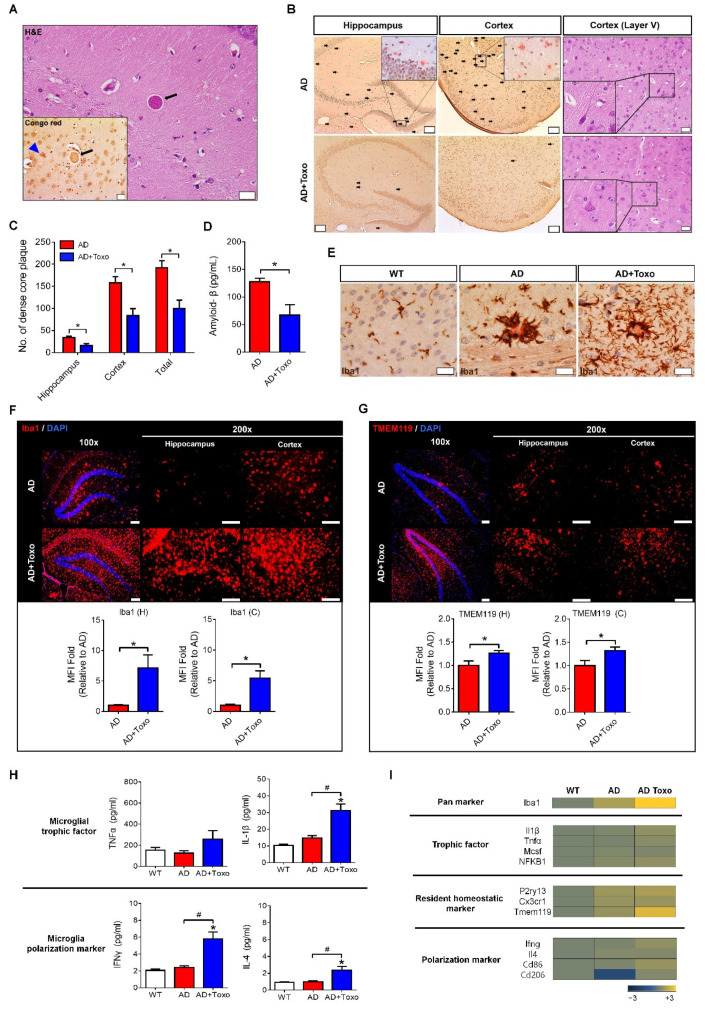
Reduction of Aβ plaques and microglial proliferation in *T. gondii*-infected 5XFAD mouse brain. (**A**) A *T. gondii* cyst (black arrow) and stained amyloid plaque (blue arrow head) in a brain section (H&E and Congo red staining) at 40 weeks PI. Scale bar; 20 µm. (**B**) Dense core plaques (black spots) stained using Congo red dye (scale bar; 100 µm) and acidophilic neurons (black quadrangle) in cortical layer V (scale bar; 20 µm). (**C**) The number of dense core plaques counted in Congo red-stained brain. (**D**) Concentration of Aβ in brain tissue lysate analyzed by ELISA. (**E**) DAB-color immunohistochemistry of microglia around Aβ plaques. Scale bar; 20 µm. (**F**) Microglial cells accumulated; Iba1-stained microglia in the brain. The fold change of MFI in the AD + Toxo group compared with the AD group in the Iba1-stained brain. Scale bar; 100 µm. H, hippocampus; C, cortex. (**G**) TMEM119-stained homeostatic microglia accumulation. The fold changes of MFI in the TMEM119-stained brain. Scale bar; 100 µm. H, hippocampus; C, cortex. (**H**) Protein expression of microglial trophic factors (IL-1β and TNF-α) and microglial polarization inducers (IFN-γ and IL-4) examined by ELISA. (**I**) Microarray analysis for microglial trophic factors (*Il1β*, *Tnfα*, *Mcsf*, and *NFKB1*), homeostatic microglial markers (*P2ry13*, *Cx3cr1*, and *Tmem119*), and inducers and markers of M1 and M2 polarized microglia (*Ifnγ*, *Il4*, *Cd86*, and *Cd206*) in AD and AD + Toxo groups compared with the gene expression in wild-type (WT) mice. Data are represented as the mean ± SEM. * Statistical significance compared with the control (* *p* < 0.05). ^#^ Statistical significance compared with each experimental group (^#^
*p* < 0.05).

**Figure 3 ijms-22-02764-f003:**
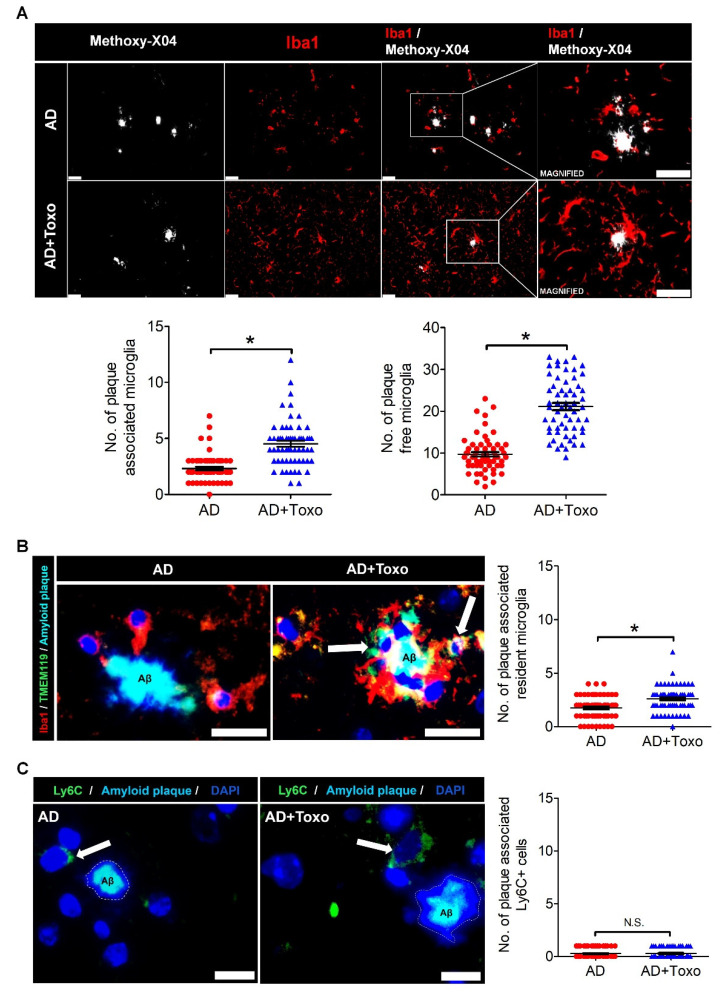
Plaque-associated patterns of microglia and Ly6C^+^ monocytes. (**A**) Iba-1-stained microglia (red) around methoxy-XO4-stained amyloid β (Aβ) plaques (white). The counting result of microglia (plaque-associated or plaque-free) found around 60 amyloid plaques distributed in the brain ((10 randomly selected plaques per mouse) × 6 mice per group). Scale bar; 25 µm. (**B**) Plaque-associated homeostatic microglia (yellow, due to co-staining of Iba-1 (red) and TMEM119 (green) and indicated by an arrow). Number of plaque-associated homeostatic microglial cells. Scale bar; 20 µm. (**C**) Ly6C^+^ monocytes (green) accumulated around Aβ plaque in brain tissues of both AD and AD + Toxo groups. (Ly6C^+^ (green) monocytes indicated by white arrow). Number of plaque-associated Ly6C^+^ monocytes. Scale bar; 20 µm. Data are represented as the mean ± SEM. * Statistical significance compared with the control (* *p* < 0.05).

**Figure 4 ijms-22-02764-f004:**
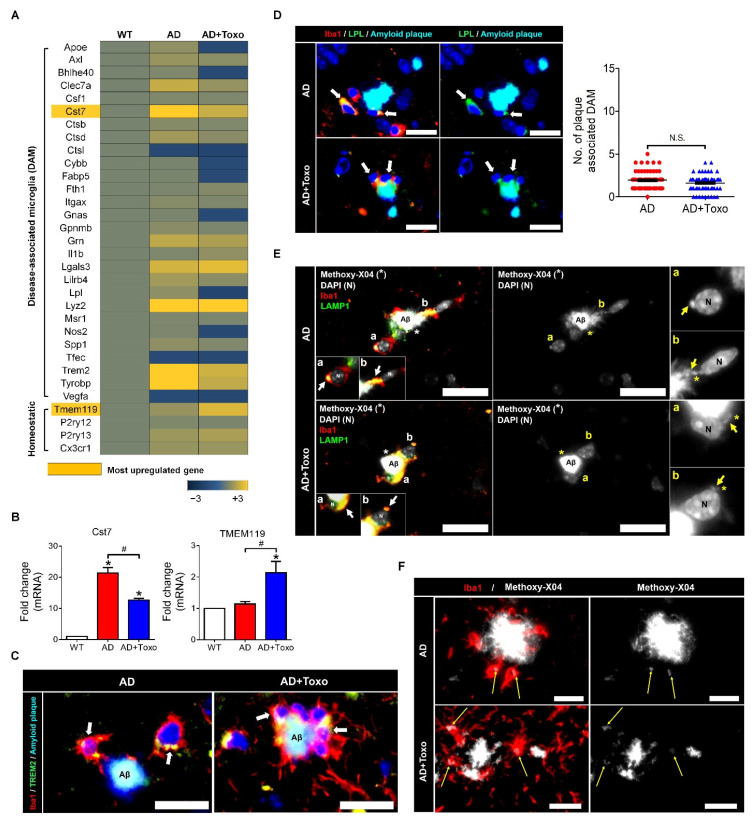
Disease-associated microglia (DAM) phenotype of microglia and microglial phagocytosis. (**A**) DAM- or homeostatic markers of microglia investigated by gene array analysis. (**B**) qPCR results for *Cst7* and *Tmem119*. (**C**) The TREM2 signal is a predictor of the DAM phenotype. (yellow, due to co-staining of Iba-1 (red) and TREM2 (green) and indicated by an arrow); Scale bar; 20 µm. (**D**) Lipoprotein lipase (LPL) signal as a stage two marker in DAM. The number of plaque-associated and LPL-expressing microglia. The counting result of microglia found around 60 amyloid plaques distributed in the brain ((10 randomly selected plaques per mouse) × 6 mice per group). (yellow, due to co-staining of Iba-1 (red) and LPL (green) and indicated by an arrow); Scale bar; 20 µm. (**E**) LAMP1 expression, and colocalization of LAMP1 and Iba1 in microglia around the amyloid β (Aβ) plaque. Lysosomal degradation of Aβ plaque as seen by the internalized puncta per microglial cell. Both “a” and “b” indicate LAMP1 positive lysosomes (white and yellow arrow) with the colocalization of methoxy-X04 stained amyloid β (*). Scale bar; 20 µm. (**F**) Internalized puncta per microglial cell indicate lysosomal degradation of the Aβ plaque (yellow arrow). Scale bar; 20 µm. Data are represented as the mean ± SEM. * Statistical significance compared with the control (* *p* < 0.05). ^#^ Statistical significance compared with each experimental group (^#^
*p* < 0.05).

**Figure 5 ijms-22-02764-f005:**
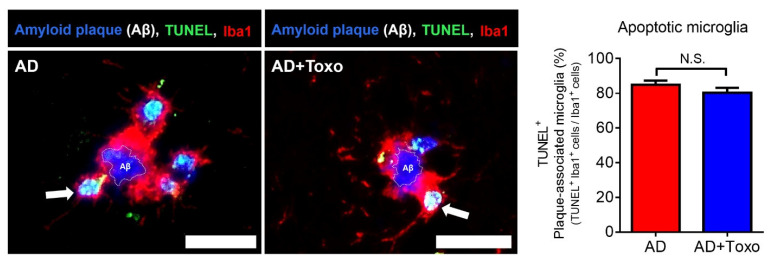
Apoptosis of plaque-associated microglia. TUNEL^+^-microglial cells surrounding the amyloid β (Aβ) plaque. Arrows represent Iba1 and TUNEL double-positive cells (light blue). Number of Iba1 and TUNEL double-positive cells in plaque-associated microglia. The counting result of plaque-associated apoptotic microglia found around 60 amyloid plaques ((randomly selected 10 plaques per mouse) × 6 mice per group). Scale bar; 20 µm. Data are represented as the mean ± SEM.

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
