# Peer review of "Reduction of Amyloid Burden by Proliferated Homeostatic Microglia in Toxoplasma gondii-Infected Alzheimer’s Disease Model Mice"

_ijms, 2021, doi:10.3390/ijms22052764_

Round 1

Reviewer 1 Report

This is a superbly executed study covering many different issues in the field of great importance.  The findings are contrary to what l might have expected but the thoroughness of the study makes me not question their results.  It had been concluded from other studies that IL4 would be increased, but these authors have clearly shown this was not the case.  The study is of high significance to the field.

The only thing missing is a couple of sentences in the discussion on how the findings of T. gondii infection can be translated to a relevant AD therapeutics.  They could also add one or two references when comparing their findings to that of Green and colleagues who have been showing the therapeutic benefits of removing the microglia.  What are the common features and what are the differences.

I see that the authors used an english service.  The manuscript is very well written but there are a couple of areas that should be examined not for grammar but to confirm the english usage was correct for the information being relayed.

Author Response

Reviewer #1: Comments and Suggestions for Authors

This is a superbly executed study covering many different issues in the field of great importance.  The findings are contrary to what l might have expected but the thoroughness of the study makes me not question their results.  It had been concluded from other studies that IL4 would be increased, but these authors have clearly shown this was not the case.  The study is of high significance to the field.

  • Thank you for reviewing our work. We also expect our study to play important roles in discovering treatments for Alzheimer’s disease. Your understanding and trust in our research will encourage us to continue with our research.

The only thing missing is a couple of sentences in the discussion on how the findings of T. gondii infection can be translated to a relevant AD therapeutics.  They could also add 1) one or two references when comparing their findings to that of Green and colleagues who have been showing the therapeutic benefits of removing the microglia.  2) What are the common features and what are the differences.

  • Thank you. We agree with your suggestion and have inserted this reference in the Discussion section (p25). Changes are in red font.
  • 1) We have included two references in the Discussion section (p25, line 8-13) regarding the findings of Green and colleagues.
  • 2) We have discussed common and different features between our findings and those of Green et al. in the Discussion section (p25, line 13-23).

I see that the authors used an english service.  The manuscript is very well written but there are a couple of areas that should be examined not for grammar but to confirm the english usage was correct for the information being relayed.

- Thank you for understanding that we are not native speakers. The manuscript was written by us, and grammar was evaluated by native speakers. We used an English editing service again. The certificate for English editing is attached to the revised version.

Reviewer 2 Report

The paper is potentially useful as foundational information for developmental treatment research in Alzheimer's disease. Major comments: The Figures are very difficult to follow and contain too much information per Figure. Figure 1 is particularly difficult. Perhaps it would be better to have more Figures with clearer Legends.

Minor comments: In Abstract line 8, amyloidosis is a different set of diseases; amyloid plaques is better. Paragraph 3 in Introduction might better begin  a separate section titled Background and Rationale; as it is the Introduction is much too long.

Author Response

Reviewer #2:

The paper is potentially useful as foundational information for developmental treatment research in Alzheimer's disease. Major comments: The Figures are very difficult to follow and contain too much information per Figure. Figure 1 is particularly difficult. Perhaps it would be better to have more Figures with clearer Legends.

  • Thank you for the encouragement.
  • We revised Figure 1 and its legend in the revised manuscript. Revised content is written in red font in the Results section and Figure Legend.

Minor comments: In Abstract line 8, amyloidosis is a different set of diseases; amyloid plaques is better. Paragraph 3 in Introduction might better begin  a separate section titled Background and Rationale; as it is the Introduction is much too long.

  • Thanks for your advice. In accordance with your suggestion, we have revised “the severity of amyloidosis” to “amyloid plaque burden” in the Abstract section. Changes are in red font.
  • You suggested including paragraph 3 in Introduction in a separate section titled Background and Rationale. We did consider including paragraph 3 in a separate section. However, the International Journal of Molecular Sciences does not stipulate the use of separate “introduction” and “background and rationale” sections. We have no choice but to follow the format in the Instructions for Authors. Please understand our situation.
  • We agree that “the introduction is much too long.” However, we have included all this information in the Introduction section because we want our readers to understand the rationale behind our study. Please understand our intention.

Round 2

Reviewer 2 Report

I think the authors have satisfied the small number of issues and the paper is ready to publish. Sorry about the communication problem. If you need the forms maybe send them to me again and I will try again. Dan Nixon